# Microdetection of Nucleocapsid Proteins via Terahertz Chemical Microscope Using Aptamers

**DOI:** 10.3390/s24227382

**Published:** 2024-11-19

**Authors:** Xue Ding, Mana Murakami, Jin Wang, Hirofumi Inoue, Toshihiko Kiwa

**Affiliations:** 1Graduate School of Interdisciplinary Science and Engineering in Health Systems, Okayama University, Okayama 700-8530, Japan; pm7g9k5d@s.okayama-u.ac.jp (X.D.); p1j130so@s.okayama-u.ac.jp (M.M.); wangjin@okayama-u.ac.jp (J.W.); 2Graduate School of Medicine Dentistry and Pharmaceutical Sciences, Okayama University Hospital, Okayama 700-8558, Japan; inoue-h1@cc.okayama-u.ac.jp

**Keywords:** terahertz chemical microscope, aptamers, N protein, microdetection

## Abstract

In the detection of the severe acute respiratory syndrome coronavirus 2 (SARS-CoV-2), several methods have been employed, including the detection of viral ribonucleic acid (RNA), nucleocapsid (N) proteins, spike proteins, and antibodies. RNA detection, primarily through polymerase chain reaction tests, targets the viral genetic material, whereas antigen tests detect N and spike proteins to identify active infections. In addition, antibody tests are performed to measure the immune response, indicating previous exposure or vaccination. Here, we used the developed terahertz chemical microscope (TCM) to detect different concentrations of N protein in solution by immobilizing aptamers on a semiconductor substrate (sensing plate) and demonstrated that the terahertz amplitude varies as the concentration of N proteins increases, exhibiting a highly linear relationship with a coefficient of determination (R^2^ = 0.9881), indicating that a quantitative measurement of N proteins is achieved. By optimizing the reaction conditions, we confirmed that the amplitude of the terahertz wave was independent of the solution volume. Consequently, trace amounts (0.5 μL) of the N protein were successfully detected, and the detection process only took 10 min. Therefore, this study is expected to develop a rapid and sensitive method for the detection and observation of the SARS-CoV-2 virus at a microdetection level. It is anticipated that this research will significantly contribute to reducing the spread of novel infectious diseases in the future.

## 1. Introduction

The existing coronaviruses include human coronaviruses (HCoVs), namely HCoV-229E, HCoV-OC43, HCoV-NL63, and HCoV-HKU1. They are widely prevalent in humans as common cold pathogens. The three main coronaviruses identified since 2002 to date, severe acute respiratory syndrome coronavirus (SARS-CoV), Middle East respiratory syndrome coronavirus (MERS-CoV), and SARS-CoV-2, have caused millions of deaths [1,2,3,4,5,6,7,8]. The most widely employed methods for confirming the diagnoses are polymerase chain reaction (PCR) and antigen testing [9,10]. PCR testing involves collecting a saliva sample or bipharyngeal swab of the patient, reverse transcribing the viral genomic ribonucleic acid (RNA) to deoxyribonucleic acid (DNA), and amplifying it [11]. This test is highly accurate; however, amplification is time-consuming and requires sample transfer. Thus, it is unsuitable for rapid testing. Contrarily, antigen detection can be completed in 15–30 min; however, it is not as accurate as PCR [12,13]. Therefore, both methods present limitations in terms of practical applications. Recently, biosensors, such as colorimetric biosensors, localized surface plasmon resonance, field-effect transistor-based biosensors, and surface-enhanced Raman scattering, have been applied with biomarker detection [14,15,16,17]. And terahertz metamaterials are also utilized for virus detection [18,19,20,21]. However, it generally lost its sensitivity for aqueous samples. Thus, we propose the application of a developed biosensor to detect proteins contained in the SARS-CoV-2, enabling the early, rapid, and highly sensitive detection of the virus. And since the sample does not directly interact with terahertz waves, the influence of aqueous solvents can be minimized.

The SARS-CoV-2 virus particle comprises four structural proteins: spike (S), nucleocapsid (N), membrane (M), and envelope (E) proteins [22]. Among them, the N protein is one of the most abundant structural proteins in virus-infected cells. Its molecular weight is 50 kDa, and the diameter and length of its filamentous nucleus are in the range of 10–15 nm and approximately 100 nm, respectively [23,24]. Importantly, it participates in RNA packaging and viral particle release. Compared with other structural proteins, N proteins are evolutionarily conserved and can be used as diagnostic markers and drug targets. Serodiagnostics shows that serum-specific antibodies against N proteins in SARS patients exhibit higher sensitivity and longer duration than other structural proteins of SARS-CoV [24,25,26,27]. Highly specific anti-N antibodies can be detected in the early stage of infection. Thus, we selected the N protein as an antigen for detection in this study.

Aptamers are DNA or RNA molecules selected in vitro by the Systematic Evolution of Ligands by Exponential Enrichment (SELEX) technology. They are functionally similar to antibodies and can bind with a wide range of molecules with high affinity and specificity [28,29]. Compared with antibodies, aptamers are widely used in basic research, drug discovery, diagnostics, and therapeutics in medicine and pharmacy because of their targeted binding ability, versatility, chemical stability, non-immunogenicity, ease of chemical synthesis, relatively small size, and low cost [30]. Thus, biotin-labeled aptamers that specifically bind to the N protein of SARS-CoV-2 were used here.

With the development of semiconductor and laser technologies, terahertz waves have recently attracted considerable attention in the biomedical field. It is challenging to apply terahertz wave technology to liquid detection because of the strong absorption property of terahertz waves in water [31,32,33,34]. Therefore, to overcome the difficulty of detection in solution, we have developed a terahertz chemical microscope (TCM) using an inductive plate as the detection element to detect biomolecules in solution [35,36,37]. Compared with PCR assays, TCM does not require pre-positioning due to amplification and sample transportation, enables the measurement of small samples in a short time, and prevents infection during transportation. Furthermore, since the assessment is performed using the numerical value of the terahertz wave amplitude, quantitative assessments can be performed without relying on the proficiency of the operator. The TCM is anticipated as a new virus detection method that is simpler in terms of preprocessing than PCR testing and more accurate than antigen detection. In addition, it is expected to significantly contribute to reducing the spread of infectious diseases.

## 2. Experimental Setup of the Terahertz Chemical Microscope

The TCM uses terahertz waves to obtain information on a sample on a semiconductor substrate called a sensing plate, which is a potential sensor [38,39]. Figure 1a shows the device structure of the TCM, and Figure 1b shows the sensing plate (10 mm^2^), comprising layers of silicon oxide (SiO_2_) films on a sapphire (Al_2_O_3_) substrate. The thickness of each layer was a few nanometers of SiO_2_, 500 nm of Si, and 500 µm of Al_2_O_3_. A sensing plate is a device that generates terahertz waves by irradiating a femtosecond laser. The generated terahertz wave amplitude is dependent on the surface potential of the sensing plate. Thus, chemical reactions, such as bonding and dissociation of samples, can be measured label-free and at high resolution by measuring changes in the terahertz wave amplitude [40,41,42]. The TCM uses a femtosecond laser (FemtoFiber ultra780, TOTOPICA Photonics AG, Munich, Germany) as the laser light source. The average output power, repetition frequency of the laser pulse, pulse width, and center wavelength were 500 mW, 80 MHz, 130 fs, and 780 nm, respectively. Using a beam splitter (SCD-500, Spectrum Detector Inc., Lake Oswego, OR, USA), the femtosecond laser was split into a pump beam for generating terahertz waves and a detection beam for detecting terahertz waves. The pump light passes through a focusing lens, irradiates the sapphire surface of the sensing plate, and generates terahertz waves proportional to the square root of the electricity on the surface of the plate. As shown in Figure 1c, the probe light is focused by the focusing lens and irradiated to excite the carriers on the detector. At this time, when the generated terahertz wave irradiates the detector, the carriers excited by the probe light beam are accelerated to generate a transient current proportional to the oscillating electric field of the terahertz wave. Considering that the instantaneous current is extremely small, it is amplified by a current amplifier (DLPCA-200, FEMTO^®^ Messtechnik GmbH, Berlin, Germany) and fed into a lock-in amplifier (LI5640, NF Circuit Design Block Co., NF Corporation, Yokohama, Japan). The lock-in amplifier obtains the terahertz waveforms by feeding the reference signals from the chopper. Synchronized detection was performed to obtain the terahertz wave field strength, which was analyzed using a personal computer. In this experiment, the sensing plate was securely attached to the measurement substrate, as illustrated in Figure 1d. The sensing plate could be uniformly divided into 4 and 9 wells. Figure 1e presents a cross-section of the measurement substrate. By changing the position of the laser irradiation on the sensing plate, the terahertz wave amplitude at different positions on the sensing plate can be measured. This enables the measurement of the response condition at any position on the sensing plate. The measurement time to obtain the amplitude at one position is approximately 300 ms, which is limited by the measurement bandwidth. However, due to the limitation of the mechanical scanning of the laser, it generally takes 10 min for 10 mm^2^ mapping of the amplitude.

## 3. Materials and Methods

Figure 2 shows the process of measuring the N protein. First, the sensing plates were ultrasonically cleaned using acetone (99.5%, Sigma-Aldrich Japan G.K., Meguro-ku, Tokyo, Japan) and ethanol (99.5%, Hayashi Pure Chemical Industries, Osaka, Japan) sequentially for 2 min to remove surface oil. After the surface oil was removed and the plates were sterilized, NaOH solution (1.0 mol/L, FUJIFILM Wako Pure Chemicals Corporation, Osaka-shi, Osaka, Japan) diluted to 200 mM with Milli-Q water was added. The plates were shaken on a Corning LSE orbital shaker (Cambridge Scientific Corp., Watertown, MA, USA) at 45 rpm for 5 min to produce hydroxyl groups on the SiO_2_ surface. After removing the NaOH solution from the wells, (3-aminopropyl) triethoxysilane (99%, Sigma-Aldrich, Saint Louis, MI, USA) diluted to 2% with Milli-Q water was added to the well. The reaction was shaken at 45 rpm for 30 min to facilitate the formation of amino groups on the SiO_2_ surface. Next, 10 mM bis(sulfosuccinimidyl)suberate (BS3) prepared in phosphate-buffered saline (PBS) (1X, Thermo Fisher Scientific, Waltham, MA, USA; pH = 7.4) was added for 1 h at 22 °C. It was left for 1 h to activate the amino groups on the SiO_2_ surface. After removing the BS3 solution and washing the surface, avidin (affinity purified, Vector Laboratories, Inc., Burlingame, CA, USA) diluted to 1 mg/mL in PBS was injected and left in a wet box at 4 °C for 24 h to crosslink the amino groups. Afterward, the avidin solution was removed and rinsed once with PBS, followed by the addition of ethanolamine-HCl (1.0 M, G from E Healthcare, Chalfont St. Giles, UK). The mixture was left at room temperature for 15 min to prevent a nonspecific reaction of the excess amino groups on the silica surface. After removing the ethanolamine-HCl from the wells and rinsing with PBS, biotin-labeled aptamers (anti-SARS-CoV-2 N-protein aptamers, purity ≥ 95%, RayBiotech., Norcross, GA, USA) diluted in PBS (100 µM, RayBiotech, Inc.) were added, and the wells were shaken for 150 min at 45 rpm on the shaker. At this point, the aptamers were immobilized on the sensing plate by the avidin–biotin binding reactions. After the reaction with the aptamers, the solution was removed and rinsed once with PBS. Finally, N protein (1.58 mg/mL, recombinant SARS-CoV-2 N protein, purity > 90%, RayBiotech) diluted with PBS was added. The aptamers reacted with the N protein by shaking at 45 rpm for 40 min on the shaker.

## 4. Results

### 4.1. Optimization of Aptamer Concentration

#### 4.1.1. Confirmation of Optimal Concentration Using the TCM

To obtain the optimal aptamer concentration for the reaction with the N protein, the sensing plate was immobilized on a four-well reaction plate. After pretreating the sensing plate, the avidin–biotin reaction was applied to immobilize the aptamers at concentrations of 5, 50, and 500 µg/mL on the sensing plate. Afterward, the measurements were performed using the TCM to obtain the terahertz wave amplitude distributions as pre-reaction data for the N proteins. After the measurement, the N protein at a concentration of 100 ng/mL was injected into each well to react with the aptamers. After the reaction, the TCM was used to obtain terahertz wave amplitude distributions as post-reaction data for the N protein. Thereafter, the amount of N protein that reacted with each concentration of aptamers was detected by comparing the terahertz wave amplitude distributions of the N protein prior to and after the reaction. Three different sensing plates were used in this experiment. Figure 3 shows an example of the obtained terahertz wave amplitude distribution. The horizontal and vertical axes represent the measurement ranges, and the different colors indicate the differences in the terahertz wave amplitude. The comparison showed that the terahertz wave amplitude changed prior to and after the N protein reaction.

Figure 3d shows the relationship between the concentration of N protein aptamers and the terahertz wave amplitude of the three sensing plates. These data were obtained by calculating the average terahertz wave amplitude for each well based on the distribution of terahertz wave amplitude prior to and after the N protein reaction and then differencing the average values. The averages were calculated from the 2 mm^2^ black box area for each well. The horizontal axis represents the aptamer concentration, and the vertical axis represents the change in terahertz wave amplitude after the N protein reaction. Even at the same aptamer concentration, the terahertz wave amplitude varied among the sensing plates because of the different nature and sensitivity of each plate. Figure 3d shows a difference in the change in terahertz wave amplitude between the aptamers at 5 µg/mL and those at 50 and 500 µg/mL, suggesting that not all the 100 ng/mL N proteins reacted with the 5 µg/mL aptamers. No difference was observed in the terahertz wave amplitude changes between the 50 and 500 µg/mL aptamers, suggesting that almost the same amount of N proteins reacted with both aptamer concentrations. These results indicate that the aptamer concentration of 50 µg/mL can detect 100 ng/mL N protein with the highest sensitivity.

#### 4.1.2. Aptamers Immobilization Confirmation via Atomic Force Microscopy

Atomic force microscopy (AFM) employs a minuscule probe to gather information about the surface topography of a sample by detecting the interactions between the probe and the atoms on the sample surface as the probe approaches. The interaction forces include Van der Waals forces and repulsive forces, which vary as the probe interacts with the sample surface. By accurately measuring these interaction forces, AFM can determine the arithmetic mean roughness (Sa) of the sample surface. In this study, we examined the surface of the sensing plate using AFM before and after the transfer of the aptamers. The observation area of the AFM was set to 10 × 10 µm. Figure 4 illustrates the changes in the surface of the sensing plate before and after aptamer immobilization. Figure 4, the arithmetic roughness increased by 1.07 nm following aptamer immobilization. This increment of roughness may be caused by the binding of small particles of aptamers on the surface.

### 4.2. N Protein Detection

#### 4.2.1. Terahertz Amplitude Variation with Solution Concentration

Based on the aforementioned experiments, we concluded that the optimal aptamer concentration was 50 µg/mL in the reaction with 100 ng/mL N protein. Under the same reaction conditions, the different detection sensitivities of the three sensing plates led to different amplitude changes in the terahertz wave. To solve this problem, we attempted to apply a sensing plate divided into four regions and conducted four sets of experiments under the same reaction conditions to verify the relationship between the change in the terahertz wave amplitude and the N protein concentration. Figure 5a shows the obtained terahertz wave amplitude distributions. The terahertz wave amplitude varied with the N protein concentration. The N protein concentrations used in the experiments were 0 (reference value), 10, 20, and 50 ng/mL. The measurement range was 8.0 × 8.0 mm. The terahertz wave amplitude varied with the N protein concentration (Figure 5a). The range of the 2.0 × 2.0 mm black box (Figure 5) was used to calculate the average terahertz wave amplitude at different concentrations for each well after the reaction. The terahertz wave amplitude at each concentration differed from the reference to afford the actual value of the terahertz wave amplitude change. Figure 5b shows the relationship between the N protein concentration and the change in terahertz wave amplitude. The horizontal and vertical axes represent the N protein concentration and terahertz wave amplitude, respectively. The error bars represent the standard deviation of the terahertz wave amplitude after four sets of measurements. Figure 5b shows the relationship between N protein concentration and the change in terahertz wave amplitude. The horizontal axis represents the N protein concentration, while the vertical axis indicates the terahertz wave amplitude. The error bars denote the standard deviation of the terahertz wave amplitude based on four sets of measurements. The results demonstrate that the terahertz wave amplitude varies as the concentration of N protein increases, exhibiting a highly linear relationship with a coefficient of determination (R^2^ = 0.9503).

#### 4.2.2. Terahertz Amplitude Variation with Solution Volume

In the aforementioned experiments, we preliminarily verified the relationship between the N protein concentration and terahertz wave amplitude. Next, we attempted to detect trace amounts of N proteins by further reducing the volume of the measurement solution to improve the usefulness of the TCM for protein detection.

A sensing plate with nine wells was used for the experiment. First, the aptamers with a concentration of 50 µg/mL were dropped into the nine wells in liquid amounts of 0.5, 1, and 2 µL (Figure 6a) for three samples in each group. The terahertz wave amplitude distribution was obtained as pre-reaction data using the TCM measurements after the reaction. After the assay, each well was reacted with the same liquid amount of the N protein solution at a concentration of 1 ng/mL. After the reaction, the terahertz wave amplitude distribution was measured using the TCM as the post-reaction data of 1 ng/mL N protein. Afterward, the aforementioned experiment was repeated to obtain data after the reaction of the N protein at concentrations of 10 and 100 ng/mL.

In this experiment, a single sensing plate was used. Figure 6a–d show the obtained terahertz wave amplitude profiles. Figure 6a shows the distribution of terahertz wave amplitude in the aptamer reaction antibody. Figure 6b–d show plots of the terahertz wave amplitude distribution after the reactions of 1, 10, and 100 ng/mL N protein, respectively. Figure 6a–d show that the terahertz wave amplitude changed prior to and after the N protein reaction. Figure 6e shows the changes in terahertz wave amplitude between different concentrations of N proteins and varying sample volumes. The results indicate the sensitivity was independent of the sample volume. The plot was generated by calculating the difference between the average terahertz wave amplitude in each 1 mm^2^ region of the holes depicted in Figure 6b–d and the average terahertz wave amplitude in each 1 mm^2^ region of the holes shown in Figure 6a. The results show that variations in terahertz wave amplitude are independent of sample volume; therefore, the three sets of sample data were analyzed using linear regression. The findings are shown in Figure 6f, where the terahertz wave amplitude demonstrates a highly linear relationship with increasing concentrations of N proteins, as evidenced by a coefficient of determination (R^2^ = 0.9881). This suggests that a quantitative measurement of N protein concentration has been successfully achieved. As shown in Figure 6, 0.5 pg of N protein was successfully detected by calculating the product of the N protein concentration and the amount reflected.

## 5. Discussion

Here, we proposed and demonstrated the application of the TCM to detect trace amounts of N protein derived from SARS-CoV-2 using aptamers. We successfully detected 0.5 pg of the N protein. To detect the N protein, we used the TCM to detect the aptamers and optimize the aptamer concentration. Furthermore, we imaged the immobilization of avidin and the aptamers by AFM and compared the root mean square values to verify the immobilization status of the aptamers. Through aptamer detection, we easily discovered evident differences in detection sensitivity between different sensing plates due to the limitations of the manufacturing industry. Thus, in the subsequent detection, the differences between the sensing plates were avoided by adjusting the detection scheme to save time and cost, and the relationship between the terahertz wave amplitude change and N protein concentration was confirmed. We confirmed that the amplitude of the terahertz wave was independent of the liquid volume during the trace detection of the N protein. The results indicated that the TCM can detect the binding state of proteins by detecting potential changes on the sensing plates. Compared with conventional detection methods, this method offers the advantages of low cost, rapid detection, and sensitivity. Therefore, the proposed method can contribute to rapid detection and virus prevention.

## Figures and Tables

**Figure 1 sensors-24-07382-f001:**
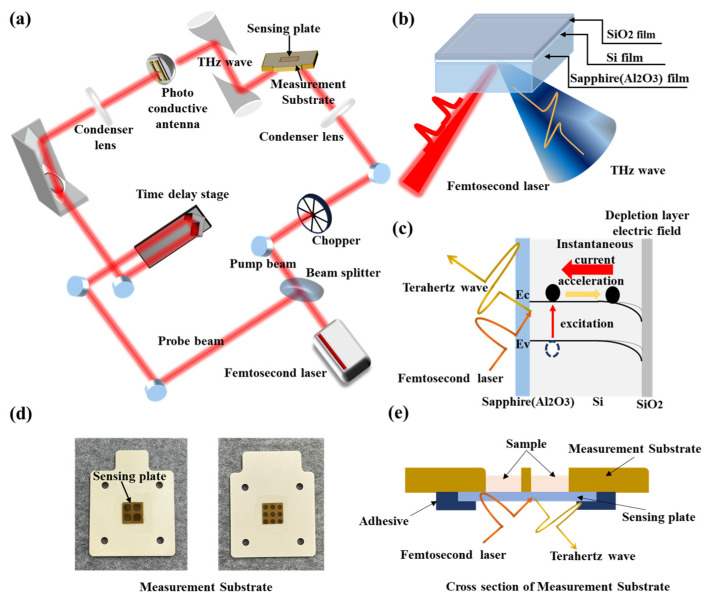
(**a**) Optical system diagram of the TCM, where the femtosecond laser is divided by a beam splitter into a pump wave that irradiates the sensing plate to generate terahertz waves and a probe beam that penetrates the photoconductor antenna for detecting the generated waves. (**b**) Schematic diagram of the sensing plate and terahertz waves radiated by the sensing plate. (**c**) Cross-section of the sensing plate used as the terahertz wave generator. When a femtosecond laser is irradiated onto the sensing plate from the sapphire film, the carriers in the silicon layer are excited and accelerated by the electric field of the depletion layer at the interface between the silicon and silicon oxide, generating an instantaneous current and a terahertz wave proportional to the time derivative of the instantaneous current. (**d**) Photograph of the sensing plate fixed to the measurement substrate. (**e**) Cross-section of the measurement substrate.

**Figure 2 sensors-24-07382-f002:**
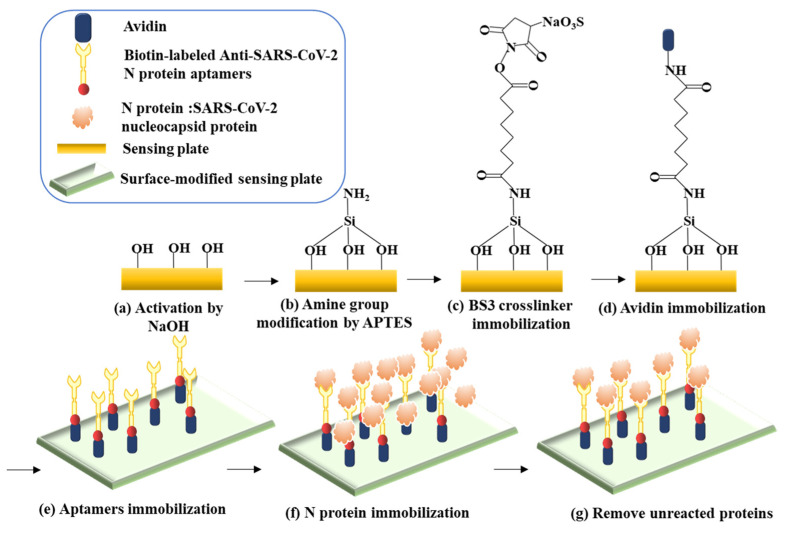
Schematic of the experimental design on the sensing plate. (**a**–**d**) Modification of SiO_2_ film. (**e**) Immobilization of aptamers by the avidin–biotin reaction; the reaction was measured using the TCM on the sensing plate to obtain the terahertz amplitude before the N protein reaction. (**f**) Addition of N protein for specific reactions with aptamers. (**g**) After removing the unreacted N protein, the reaction was measured using the TCM to obtain the terahertz amplitude after the N protein reaction.

**Figure 3 sensors-24-07382-f003:**
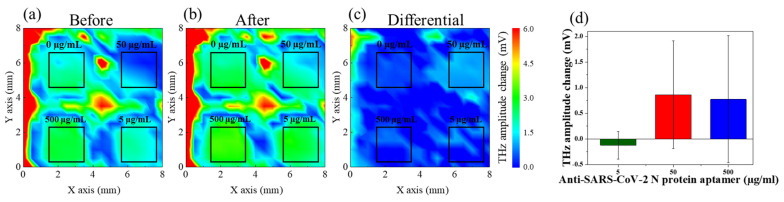
(**a**) Distribution of terahertz amplitudes on the sensing plate after the immobilization of the aptamers. (**b**) Distribution of terahertz amplitudes on the sensing plate after the N protein reaction. (**c**) Terahertz wave amplitude changes on the sensing plate prior to and after the N protein reaction. (**d**) The change in terahertz wave amplitude following the reaction of N protein at a concentration of 100 ng/mL with aptamers at concentrations of 5, 50, and 500 µg/mL, respectively.

**Figure 4 sensors-24-07382-f004:**
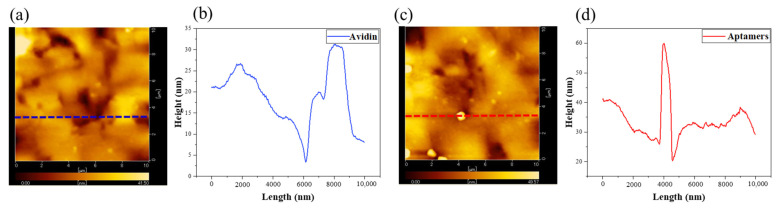
Surface conditions and height profiles after avidin and aptamers binding to sensing plates via AFM. (**a**,**b**) Overall surface condition and height profile after avidin; the arithmetic mean roughness (Sa) was approximately 4.79 nm. (**c**,**d**) Overall surface condition and height profile after aptamers; the Sa was approximately 5.86 nm.

**Figure 5 sensors-24-07382-f005:**
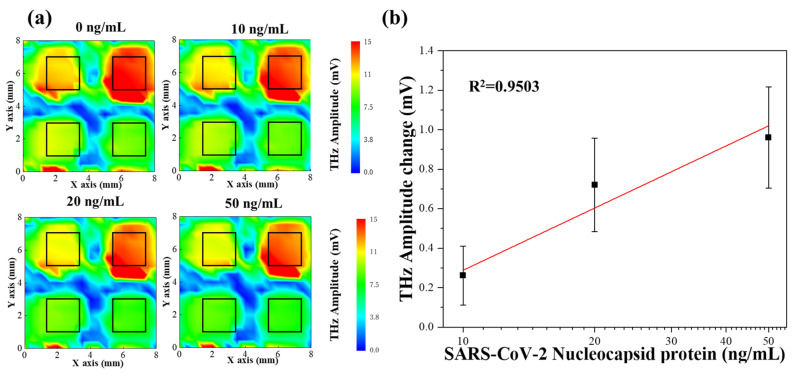
(**a**) Imaging after aptamer interaction with N protein at different concentrations (0–50 ng/mL) via TCM. The area of the black squares (2.0 × 2.0 mm) was used to calculate the average terahertz amplitude. The amplitude of the terahertz wave (color intensity) changed with a change in the N protein concentration. Contour images were obtained using Origin 2020 software. (**b**) Quantitative analysis of interactions between the aptamers and N protein. The error bars indicate the standard deviation (*n* = 4).

**Figure 6 sensors-24-07382-f006:**
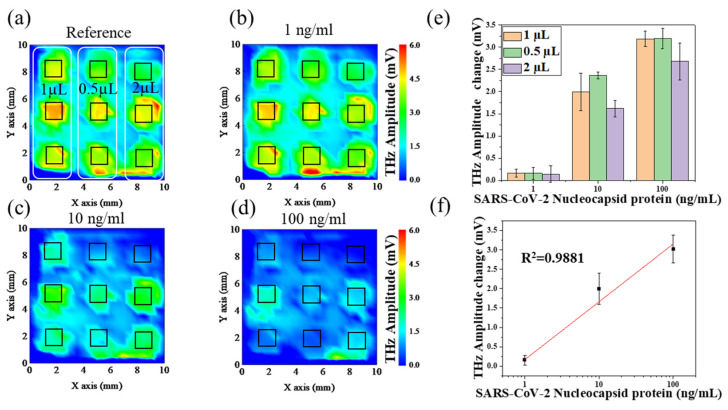
(**a**–**d**) Imaging after aptamer interaction with N protein at different concentrations (0–100 ng/mL) via TCM. The area of the black squares (1.0 × 1.0 mm) was used to calculate the average terahertz amplitude. The amplitude of the terahertz wave (color intensity) changed with a change in the N protein concentration. Contour images were obtained using Origin 2020 software. (**e**) The variations in terahertz wave amplitude across different concentrations of the N protein at varying sample volumes. (**f**) A quantitative analysis of the interactions between aptamers and the N protein. Error bars represent the standard deviation (*n* = 9).

## Data Availability

The data that support the findings of this study are available from the corresponding author, [T.K.], upon reasonable request.

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
