# Peer review of "Microdetection of Nucleocapsid Proteins via Terahertz Chemical Microscope Using Aptamers"

_sensors, 2024, doi:10.3390/s24227382_

Round 1
Reviewer 1 Report
Comments and Suggestions for Authors
I think this paper is interesting for researchers of sensor devices. You would better to revise some parts to improve it.
1. The well structure is not clear. An explanation should be added to Fig.1 or in the text.
2. In section 4.1.2, the heights prior to and after the immobilization of the aptamers are written. However, it is not sure what these results values mean. You should write comments.
3. The area of the black box size was used to calculate the average terahertz amplitude. However, the size in the figure does not match the size of the text. For example, they are1.5mm2 in Fig. 3, 2.0x2.0mm in Fig.5, and 1.0x1.0mm in Fig.6. The size of the black box in all the figures seem to be same. You should check these values.
4. Inspection time is estimated at ten minutes, but the back grounds of this value are not explained. Estimation of this value should be written using such as the laser scan scanning speed, the size of the scanning range, numbers of measurement points or etc.
By addressing these points, the clarity and overall impact of the paper could be significantly improved for the target audience.
Author Response
We wish to express our appreciation to the Reviewer for his or her insightful comments, which have helped us significantly improve the paper.
Comment 1: The well structure is not clear. An explanation should be added to Fig.1 or in the text.
Response:
Thank you very much for your suggestion. It was an oversight on my part not to explain it clearly. I have added diagrams (d) and (e) to Figure 1 and provided explanations for them in lines 110-113 of the article as follows
“In this experiment, the sensing plate was securely attached to the measurement substrate, as illustrated in Figure 1(d). The sensing plate could be uniformly divided into 4 and 9 wells. Figure 1(e) presents a cross section of the measurement substrate.”
Comment 2: In section 4.1.2, the heights prior to and after the immobilization of the aptamers are written. However, it is not sure what these results values mean. You should write comments.
Response:
Thank you very much for your suggestion. I have revised the article and explained the meaning of the result values in lines 218-221 as follows
“Figure 4 illustrates the changes in the surface of the sensing plate before and after aptamers immobilization. Figure 4, the arithmetic roughness increased by 1.07 nm following aptamers immobilization. This increment of roughness may cause by binding of small particles of aptmers on the surface.”
Comment 3: The area of the black box size was used to calculate the average terahertz amplitude. However, the size in the figure does not match the size of the text. For example, they are1.5mm2 in Fig. 3, 2.0x2.0mm in Fig.5, and 1.0x1.0mm in Fig.6. The size of the black box in all the figures seem to be same. You should check these values.
Response:
Thank you very much for your careful observation. There is no issue with the values; the problem arose from my oversight in creating the image. I made a change at line 193 as follows
“1.5-mm2.”
→“2-mm2”
In addition, I changed the dimensions of the black box in Figure 6 to align with the calculated values. The size of the black box is 1-mm2. The varying range of values is illustrated in the newly added Figure 1(d), which differs due to the varying sizes of the wells.
Comment 4: Inspection time is estimated at ten minutes, but the back grounds of this value are not explained. Estimation of this value should be written using such as the laser scan scanning speed, the size of the scanning range, numbers of measurement points or etc.
Response:
In order to clarify the issue of detection time, an explanation has been added in lines 116-118 as follows
“Measurement time to obtain the amplitude at one position is approximately 300 ms, which is limited by the measurement bandwidth. However, due to the limitation of the mechanical scanning of laser, it generally takes 10 min for 10mm mapping of the amplitude.”

Reviewer 2 Report
Comments and Suggestions for Authors
The authors proposed the application of a developed biosensor to detect proteins contained in the SARS-CoV-2, enabling the early, rapid, and highly sensitive detection of the virus. This work presents a non-invasive and innovative method that can be employed to detect nucleocapsid proteins using aptamers in conjunction with THz chemical microscopy. The combination of these technologies for chemical analysis, and micro-detection offers a potentially real-time diagnostic tool that is highly sensitive and could significantly improve the detection of viral infections, (in this case Covid-19) offering benefits such as high specificity, minimal sample preparation, and rapid results. This study has ably demonstrated how this can be made possible.
The methodology described is spot on and clear.
However, the discussion is very shallow and could have been expounded on more. The authors should compare with other studies that have attempted similar work and show how they have improved or innovated on their ideas.
Author Response
We would like to thank the reviewer for their insightful comments, which have helped us improve the paper significantly.
Reviewer Comment: However, the discussion is quite superficial and could be expanded with a more in-depth explanation. The authors should compare their work with other similar studies and demonstrate how they have improved or innovated their approach.
Response:
Theoretically, TCM can measure concentration as long as the sample volume is sufficient to cover the laser spot. However, there have been no reports indicating that sensitivity is independent of sample volume. Here, we report experimental results for the first time, showing that sensitivity is indeed volume-independent and demonstrating high sensitivity in detecting the N protein. To clarify our work, we have added the following sentence in lines 291-292:
"The results indicate that sensitivity is independent of sample volume."

Reviewer 3 Report
Comments and Suggestions for Authors
In this paper, authors detect different concentrations of N protein derived from SARS-CoV-2 by using their terahertz chemical microscope (TCM). Specially, trace amounts of N protein are detected using aptamers during this paper. The optimal aptamers concentration was obtained by using TCM, showing that 50μg/ml aptamers can detect 100ng/ml N protein. Although the methods presented in this paper have shown some advantages, there are some suggestions may be useful for improving the article.
1. Except for a few self cited articles, other references are not recent enough. This is not conducive to assessing the innovation and effectiveness of this article.
2. In addition, some terahertz sensors, such as terahertz metasurfaces, have been used for virus detection. Quoting some relevant references and making appropriate comparisons may be beneficial.
3. Can this method be used for quantitative analysis of N protein? Is there a problem of concentration saturation?
4. For verifying the conclusion of rapid detection and trace amounts, it may be considered to compare the results with existing references.
Author Response
We want to thank the reviewer for their insightful comments, which have greatly helped us improve the paper.
Comment 1: Aside from a few self-citations, most of the references are outdated, which does not support the evaluation of this paper’s innovation and effectiveness.
Response: I agree with your point, and I have added a reference on the application of terahertz technology in virus detection in line 45. Additionally, I have explained the advantages of traditional Chinese medicine in virus detection in lines 45-49 as follows:
"Terahertz metamaterials have also been used in virus detection [18-21]. However, they often lose sensitivity in aqueous samples. Therefore, we propose using the developed biosensor to detect proteins in SARS-CoV-2, enabling rapid, highly sensitive early virus detection. Since the sample does not interact directly with terahertz waves, the impact of aqueous solvents is minimized."
Comment 2: Additionally, some terahertz sensors, such as terahertz metasurfaces, have already been used in virus detection. Citing some relevant references and making an appropriate comparison may be helpful.
Response: Thank you very much for this suggestion; it has been very helpful in revising the paper. As indicated in the response to Comment 1, changes have been made based on your recommendation.
Comment 3: Can this method be used for quantitative analysis of the N protein? Is there an issue with concentration saturation?
Response: This is an excellent question. We are currently preparing a calibration curve for quantitative measurements. To address saturation issues, we are also developing a voltage-feedback sensing platform.
Comment 4: To validate the conclusions on rapid detection and trace analysis, consider comparing the results with existing literature.
Response: Thank you for your question. Commercial rapid antigen testing requires 30 minutes with a sample volume of 200 µl, whereas our system only takes 300 milliseconds to measure a sample of 0.5 µl.
